# Elevated Pediatric Chagas Disease Burden Complicated by Concomitant Intestinal Parasites and Malnutrition in El Salvador

**DOI:** 10.3390/tropicalmed6020072

**Published:** 2021-05-07

**Authors:** Melissa S. Nolan, Kristy O. Murray, Rojelio Mejia, Peter J. Hotez, Maria Jose Villar Mondragon, Stanley Rodriguez, Jose Ricardo Palacios, William Ernesto Murcia Contreras, M. Katie Lynn, Myriam E. Torres, Maria Carlota Monroy Escobar

**Affiliations:** 1Department of Epidemiology and Biostatistics, Arnold School of Public Health, University of South Carolina, Columbia, SC 29208, USA; lynnmk@mailbox.sc.edu (M.K.L.); metorres@mailbox.sc.edu (M.E.T.); 2Department of Pediatrics, Section of Tropical Medicine, National School of Tropical Medicine, Texas Children’s Hospital and Baylor College of Medicine, Houston, TX 77030, USA; kmurray@bcm.edu (K.O.M.); rojelio.mejia@bcm.edu (R.M.); hotez@bcm.edu (P.J.H.); mariajose.villarmondragon@bcm.edu (M.J.V.M.); 3Center of Health Investigation and Discovery (CENSALUD), University of El Salvador, San Salvador, El Salvador; stanleyfree82@gmail.com (S.R.); jose.palacios4@ues.edu.sv (J.R.P.); 4Basic Integrated Health System (SIBASI)-Sonsonate, Western Region, El Salvador Ministry of Health, Sonzacate, El Salvador; sunnnkkyy@yahoo.es; 5Laboratory of Applied Entomology and Parasitology, School of Biology, University of San Carlos, Guatemala City, Guatemala; mcarlotamonroy@gmail.com

**Keywords:** Chagas disease, *Trypanosoma cruzi*, pediatric, polyparasitism, intestinal parasites, malnutrition, food insecurity, El Salvador, Central America

## Abstract

The eradication of the vector *Rhodnius prolixus* from Central America was heralded as a victory for controlling transmission of *Trypanosoma cruzi*, the parasite that causes Chagas disease. While public health officials believed this milestone achievement would effectively eliminate Chagas disease, case reports of acute vector transmission began amassing within a few years. This investigation employed a cross-sectional serosurvey of children either presenting with fever for clinical care or children living in homes with known triatomine presence in the state of Sonsonate, El Salvador. Over the 2018 calendar year, a 2.3% Chagas disease seroprevalence among children with hotspot clustering in Nahuizalco was identified. Positive serology was significantly associated with dogs in the home, older participant age, and a higher number of children in the home by multivariate regression. Concomitant intestinal parasitic infection was noted in a subset of studied children; 60% having at least one intestinal parasite and 15% having two or more concomitant infections. Concomitant parasitic infection was statistically associated with an overall higher parasitic load detected in stool by qPCR. Lastly, a four-fold higher burden of stunting was identified in the cohort compared to the national average, with four-fifths of mothers reporting severe food insecurity. This study highlights that polyparasitism is common, and a systems-based approach is warranted when treating Chagas disease seropositive children.

## 1. Introduction

Infection with *Trypanosoma cruzi* parasite results in a chronic lifelong illness, called Chagas disease, that causes cardiac and intestinal organomegalies in a subset of patients. In endemic Latin American countries, infections traditionally occur in youth with chronic clinical manifestations presenting later in life. Acute vector-borne infection can be indicated by appearance of a chagoma or Romaña’s sign, yet most acute infections are believed to be asymptomatic or present with general flu-like illnesses that self-resolve. Several decades later, irreversible clinical manifestations typically present, resulting in serious personal health and societal economic impacts [1,2]. Sadly, less than 1% of Chagas disease patients receive lifesaving antiparasitic medication [3,4], most often due to lack of infection status knowledge and/or lack of healthcare access [5]. Further, timely treatment is imperative as therapeutic efficacy diminishes with disease progression. Several studies have indicated that children respond best to treatment as evidenced by higher percentages of serologic titer reductions and lower treatment discontinuation rates [6].

Chagas disease is endemic throughout the Americas, with an estimated 6.5 million infected according to the 2019 Global Burden of Disease study [7], and incidence is thought to be declining due to active vector control programs. El Salvador has long been an endemic country for Chagas disease with the first case identified in 1913 [8]. *Rhodnius prolixus* was the dominant domestic vector species associated with the majority of transmissions, and its eradication in June 2010 was believed to hold great promise for halting vector-borne transmission countrywide [9]. Despite this public health success, prospective investigations revealed a high incidence of acute Chagas disease [10,11,12,13]. The last national prevalence investigations concluded in 2011 [12], leaving a gap in current burden estimates as a new domestic vector species has since established a stronghold in the region [14,15]. Further, important clinical comorbidities are rarely studied in the context of pediatric Chagas disease. This cross-sectional study aims to develop a foundation for prospective, comprehensive investigations. To address these knowledge gaps, a pediatric *T. cruzi* seroprevalence investigation with contemporary study arms assessing intestinal parasite concomitant infection and malnutrition was conducted in the state of Sonsonate, El Salvador.

## 2. Materials and Methods

### 2.1. Study Site and Population Characteristics

The study took place in the state of Sonsonate, El Salvador. Sonsonate is located in the western region of El Salvador, close to the Guatemala border, and abutted by the Pacific Ocean. The state comprises 16 municipalities, which are then further classified into a number of cantones (translates to “counties” in English). Sonsonate is home to an estimated 520,000 persons, with a population density of 425 persons/km^2^ and an annual +0.87% population growth [16]. Sonsonate’s population demographic characteristics include that 53% are female, 60% of the population live in an urban setting, and 82% of residents are literate [16]. El Salvador as a country is dominated by younger persons, with 26% of the population being 14 years old or younger and 19% being aged 15 to 24 years old [17]. However, the country’s average maternal age at first birth is 21 years old, with an average of two children born per woman [17]. The country’s crude death rate is 7 per 1000 persons with the following top five causes of mortality: (1) diseases of the circulatory system, (2) external causes (predominately assault), (3) neoplasms, (4) disorders of the genitourinary system (predominately Mesoamerican nephropathy), and (5) infectious/parasitic diseases [18].

### 2.2. Study Recruitment and Activities

Children were recruited by local study nurses in collaboration with trusted community promotoras from two possible avenues: (1) living in a home known to the local vector control agency to either be currently or recently infested with triatomines or (2) presenting to a local clinic with acute febrile disease of unknown etiology. Location of clinic and home recruitment rotated each week to ensure all 16 municipalities were equally represented. Inclusion criteria was anyone aged 9 months to 18 years, and living within the state of Sonsonate, El Salvador at the time of enrollment. The only exclusion criteria were if a parent or legal guardian was not accompanying the child to provide consent. Enrollment occurred throughout the 2018 calendar year; however, enrollment was suspended in the first two weeks of January, the last week of March, and last week of December in respect for the Easter and Christmas holiday seasons.

The primary investigation focused on Chagas disease seroprevalence and associated risk factors; therefore, all children enrolled throughout the 2018 study period were asked to provide a serum sample for *T. cruzi* antibody testing and their parent was asked to complete a household risk factor questionnaire. All participant’s sera were tested by two different assays: Chagas Stat-pak (Chembio Diagnostic Systems Inc., Medford, NY, USA) and Hemagen Chagas Kit (Hemagen Diagnostics, Columbia, MD, USA). Chagas Detect Plus (InBios, Seattle, WA, USA) was used as a tertiary diagnostic assay for indeterminate results. The risk factor questionnaire was developed in collaboration with local experts from the University of El Salvador, SIBASI (Sonsonate health department), the University of San Carlos, and was adapted from a published Central American Chagas household risk factor survey [19].

Two different secondary pilot investigative arms were performed during select months. Anyone enrolled during these time frames was permitted to accept or decline participation in the secondary pilot investigation while still participating in the original Chagas investigation. From January to May 2018, participants were invited to take part in intestinal parasite surveillance. Willing participants were asked to provide a fecal sample in a sterile conical, which was immediately aliquoted into 1.5 mL cryovials and frozen at −20 °C for a maximum of 1 month or until samples could be transferred to a −80 °C freezer. All samples were tested in April 2020 by multi-parallel qPCR using previously published methods for the following nine intestinal parasites: *Ascaris lumbricoides*, *Ancylostoma duodenale*, *Necator americanus*, *Strongyloides stercoralis*, *Trichuris trichiura*, *Giardia lamblia*, *Cryptosporidium* species, *Entamoeba histolytica*, and *Blastocystis hominis* [20]. From September to December 2018, pediatric participants’ parent or legal guardian was asked to complete an additional food intake frequency nutritional survey. This nutritional survey was adapted from a published, validated Colombian one-week recall household food survey [21]. This nutritional survey asked about adherence to El Salvador Ministry of Health national guidelines [22], daily dietary composition, food insecurity, home hygienic practices, and self-reported health outcome indicators.

### 2.3. Ethics Statement

Human subject ethics approvals were acquired from Institutional Review Boards at Baylor College of Medicine and the University of South Carolina and from the El Salvadorian Ministry of Health. Signed parental consent and verbal child assent were collected by a study nurse prior to participation in study activities or sample collection. All approached participants and parents were provided a non-monetary (food) appreciation gift for their consideration. All children who tested positive for either Chagas disease or intestinal parasites as a part of the study’s surveillance activities were provided respective treatment at no cost through the local health department.

### 2.4. Data Analysis

Univariate logistic regression was performed to assess risk factors associated with Chagas disease positive status, using Stata SE v15.1 (Stata Corporation, College Station, TX). Post-estimate regression diagnostics were run to assess co-linearity and mean variance inflation factor was 4.09, indicating no univariate variable co-linearity. Multivariate backward stepwise model was performed for Chagas disease variable correlation analysis. Any univariate variables with *p* < 0.05 went into the multivariate model and final significance for the multivariate model was considered if *p* < 0.05. The temporality of Chagas positive case diagnoses by epidemiological week were assessed by logistic regression. Hotspot cluster analysis using Global Moran’s I statistic was executed in ArcMap 10.6.1 (ESRI Corporation, Redlands, CA, USA) to evaluate the spatial autocorrelation of the Chagas disease pediatric burden in Sonsonate state. Logistic regression identified rural versus urban living to be a better dependent indicator of nutritional status than Chagas disease status (only 3 out of 154 nutritional survey respondents were Chagas seropositive); therefore, nutritional status and correlates were assessed using univariate logistic regression, stratified by living in a rural versus urban environment.

## 3. Results

Throughout the calendar year 2018, a total of 1074 children provided serologic samples for *T. cruzi* antibody testing and had a parent complete the household risk factor questionnaire. From January to May 2018, 168 children additionally provided a fecal sample for intestinal parasite testing. From September to December 2018, 154 parents completed an additional nutritional survey. Eligibility versus enrollment and overall investigative participant workflow results are listed in Figure 1. Overall enrollment was high; however, participation in fecal sample submission was low (30%) due to disinterest in collecting fecal specimens by caretakers.

From the 2018 calendar year, a 2.3% (n = 25) Chagas seroprevalence was identified in children from the state of Sonsonate, El Salvador, defined as testing positive on a minimum of two different serologic antibody assays. False positives were common, with an additional 4.6% (n = 49) of children testing positive on at least one assay. Overall diagnostic assay performances were: 5.6% (n = 60) positive on Hemagen Chagas Kit, 2.2% (n = 24) positive on Chagas Stat-pak, and notable diagnostic discordance was noted, with Chagas Detect Plus serving as the second confirmatory assay on 68% of the true positive cases (n = 17 of 25). As demonstrated in Figure 2, *T. cruzi* antibody-positive cases were identified throughout the calendar year. Logistic regression of cases by epidemiological week was not statistically significant, suggesting a lack of seasonality in identification of seropositive children. As demonstrated in Figure 3, a significant clustering was noted, with the highest pediatric Chagas case burden originating in Nahuizalco (72% of all seropositive cases originated in this municipality).

Several risk factors were associated with seropositive children by univariate analysis (Table 1). The only demographic risk factor identified was advanced age, with older children having slightly higher increased odds (OR = 1.10, *p*-value < 0.001). The youngest seropositive child was 5 years old, with a median age of 13 years and oldest seropositive child was 18 years old. No difference was noted in seropositivity between the two originating participant populations: asymptomatic living in a known triatomine-infested home versus febrile presenting to a local clinic. Overall, the pediatric population originated from health-compromised homes, with 57% of respondents reporting at least one household member requiring clinical care at a local clinic or hospital for an infectious disease. Healthcare-seeking behaviors and history of antiparasitic medicine were not associated with Chagas disease seropositivity, however.

Larger households with more children than adults were significantly more likely to have a seropositive child. Households with more household members and more children were significantly more likely to have a seropositive child. Specifically, homes with six or more members had 2.54 times greater odds (*p*-value: 0.024; 95% CI: 1.13 to 5.72). Homes with a transient member (predominately a male relative) neared significance (*p*-value = 0.053) for increased odds of a seropositive child. A home with four or more children had 3.33 times greater odds (*p* = 0.004; 95% CI: 1.47 to 7.53) of having a seropositive child. Interestingly, number of currently pregnant women, primary income source, and household receiving “remesas” (money) from USA-based relatives were not associated with the presence of a seropositive child. Note, the primary income source was compared by both multinomial logistic of income categories (agriculture, fishing, security, commercial, and business) and by univariate logistic of agriculture versus all others (65% of respondents reported agriculture as the primary source); both calculations yielded non-significant findings. An additional household risk factor included 3.41 times greater odds of infection (*p*-value = 0.016) among homes owning dogs.

Lower maternal education was significantly associated (*p* = 0.005) with a child being seropositive. Neither completing primary school (*p* = 0.372) or completing secondary school (*p* = 0.450) were associated with seropositivity; however, a mother completing three years or less of school had 2.75 (95% CI: 1.23 to 6.12) times greater odds of a child testing seropositive (*p* = 0.013). Similarly, lower paternal education was significantly associated (*p* = 0.037) with a child being seropositive; however, households with a father who did not complete primary school had 4.76 (95% CI: 1.62 to 13.97) greater odds of a seropositive child (*p* < 0.004). Chagas disease knowledge was low among the entire cohort, but not significantly associated with infection status. Less than 20% of the entire cohort had knowledge of Chagas disease, despite 68% reporting “chinche” (triatomine vector) knowledge and 20% of those with chinche knowledge reporting a history of a family member being bitten by a chinche within the past year.

A multivariate model was performed that yielded three significant variables. Older children had increased odds of infection (*p* = 0.002; OR = 1.15, 95% CI: 1.05 to 1.26). A larger number of children in the home was associated with Chagas disease (*p* = 0.021; OR = 1.27, 95% CI: 1.03 to 1.57). A household having a dog had 2.94 (95% CI: 1.08 to 8.05) greater odds of infection (*p* = 0.035) than those households without a dog. The small number of variables that stayed significant in the final model could potentially be due to a lack of power stemming from the relatively low number of pediatric Chagas disease cases (n = 25) or the population’s homogeneity in demographics, household composition, poverty indicators, and triatomine vector exposures.

Intestinal parasites were prevalent in the pediatric cohort subset (n = 168). In total, 60% of the pediatric subset was singularly infected with the species breakdown as follows: *Trichuris trichiura* (1%), *Necator americanus* (1%), *Giardia lamblia* (4%), or *Blastocystis hominis* (54%). A notable proportion (15%) of the pediatric subset had polyparasitism with two or more intestinal parasites. *Blastocystis hominis* and *Giardia lamblia* were the most common etiologies for co-infection (Figure 4). Two children with chronic Chagas disease (*T. cruzi*) had co-infections with *Giardia lamblia* (n = 1) and *Blastocystis hominis* (n = 1). Polyparasitism was significantly associated (*p* < 0.002) with increased loads of *Giardia lamblia* and *Blastocystis hominis*. Interestingly, no children in the subset had detectable *Ascaris lumbricoides*, *Strongyloides stercoralis*, *Ancylostoma duodenale*, *Entamoeba histolytica*, or *Cryptosporidium* species infections. Parasitic infection status significantly varied (*p* < 0.002) by geographic location, and Nahuizlco had the highest burden of polyparasitism (Figure 3).

Nutrition deficiencies and insecurity were common among the second cohort subset studied. Of 154 families interviewed, 88% of families reported worrying about where their next meal would come, and 87% of children reported eating one meal more or less per day. Of the entire subset, 38% of children less than 5 years old met the WHO definition for stunted growth (N = 20/53), compared to 9% nationally and 15% regionally [22]. Approximately 61% of households lacked dietary diversity, with beans, grains, sugar, and oil being the most frequently eaten food types (Figure 5). It was anecdotally noted that a number of households reported having a cast-iron bowl of beans in the middle of the room that served as the only meal source. Additionally, bean and cheese pupusas made of rice flour and topped with curtido (a vinegar cabbage slaw mix) were commonly consumed meals. Meat was scarcely consumed, with less than 5% of households reporting any meat of any kind on their weekly food frequency recall nutritional survey. Univariate analysis stratified by rural versus urban geographic location found very few statistical differences in diet composition, food insecurity, and hygienic practices (Figure 5 and Table 2).

## 4. Discussion

This investigation revealed a foci of pediatric Chagas disease presented in a contemporary context of comorbid clinical concerns. Not only was a clustering of ongoing *T. cruzi* transmission identified, high rates of polyparasitism and malnutrition were also identified that have important implications for these children’s long-term health. In fact, children with polyparasitism had significantly higher parasite burdens on cross-sectional examination, which warrants further investigation into children’s responses following antiparasitic medications. Further, the grave rate of malnutrition and stunting already present in those 5 years and younger likely plays an important role in this population’s host immunologic response and susceptibility to infections.

Chagas disease has long plagued El Salvadorians, primarily attributable to the ancestral *Triatoma dimidiata* vector species and the imported *Rhodnius prolixus* species [23]. The 1990–2000s Pan American Health Organization’s “Iniciativa de los Países de Centro América para la Interrupción de la Transmisión Vectorial, Transfusional y Atención Médica de la Enfermedad de Chagas” resulted in the certified elimination of the leading domestic *Rhodnius prolixus* vector in El Salvador [9], leading many to believe incident cases would be negligible in the near future. *Triatoma dimidiata* was a previously overlooked species of transmission importance that has since taken *Rhodius prolixus*’s place in Central American homes [19,24]. However, *Triatoma dimidiata* is a dynamic vector that can genetically and morphologically adapt to its anthropogenic environment [14]. While the species can exist between altitudes of 300 to 1400 m [15,25], presence of this species around 700 m has been recommended for targeted vector abatement due to the vector species optimal survival and *T. cruzi* infectivity at this elevation [25]. In parallel with these studies, Nahuizalco was identified as the epicenter of pediatric Chagas disease in this investigation, which has an elevation of approximately 600 meters. While domestic vector species were not collected as part of this study, it has been well described that *Triatoma dimidiata* is present in this region [14,24,26], and the authors hypothesize this is the vector principally responsible for the pediatric infections in this population. Fortunately, recent community-focused educational programs, housing material improvement, and animal management efforts have been shown to be effective at reducing *Triatoma dimidiata* from residences in Central America [27,28,29,30], and these efforts are warranted in Nahuizalco.

These findings suggest an increase in pediatric Chagas disease prevalence over the past decade. This serosurvey identified 2.3% of children were reactive on two different antibody assays. The last surveillance efforts were performed in 2008–2011: 1% prevalence in children <16 years [31], 4% prevalence among pregnant mothers with a 3% congenital transmission rate [11], and a total of 731 acute cases continually reported from 2000–2012 [10]. In comparison, an El Salvadorian blood bank surveillance study of adults found a continual decline in annual positivity from 3.7% to 1.7% during 2001–2011 [12]. The finding of 2.3% infection prevalence among children sharply contrasts the previous decade’s Chagas disease burden, with a potential doubling in this population. A recent Honduran national effort effectively identified hotspots for targeted vector control using school-based surveillance [32], and the findings of infection in children aged 5 years and older suggest that this same culturally-accepted and resource-feasible method would be effective in Sonsonate and surrounding regions. The low number of Chagas disease seropositive children was a potential limitation, and future studies should validate these findings. The transmission source of these chronic pediatric Chagas disease cases is believed to be vectorial; however, the potential for mother-to-child transmission cannot be ruled out. This study was unable to serologically test mothers for Chagas disease, and the latest estimates suggest approximately 3% of infected mothers will congenitally transmit infection to their infants [11]. This study was also unable to establish seasonality related to vector activity since seroprevalence does not indicate acute infection. Considering the hyperendemicity of domestic triatomine species in this region [19,31], the high prevalence of acute pediatric cases diagnosed by the Ministry of Health [10], and the study’s finding of dog species present in home increasing risk (as a known risk factor for vector-human transmission cycles [24,33,34]), one would determine that a mixture of both transmission sources is responsible. As previously highlighted [11,35], the need for Chagas disease surveillance in pediatrics and women of childbearing age is warranted. The low number of Chagas disease seropositive children was a potential limitation, and future studies should validate these findings.

Malnutrition and food insecurity during childhood have significant long-term health consequences, and co-existing infectious diseases can exacerbate these health outcomes [36]. Despite the commonality of protein-energy and micronutrient malnutrition in Central America [37], only one study to date has assessed the impact of this co-occurring condition in Chagas disease seropositive humans. The combination of Chagas disease and malnutrition can result in a 2.4-fold increase of stunting compared to seronegative-matched Brazilian school-aged children [38]. Animal studies have found higher parasitemia and mortality among protein-restricted diet feed mice and impaired inflammatory immune response (lower macrophage activation of TNF-*a*, IL-10, and NO) [39]. While the study was not able to directly assess concomitant Chagas disease and malnutrition, this study did identify considerable protein and micronutrient diet insufficiencies in participants. Meat consumption was a scarcity, and most families ate one meal a day which consisted of the same food item. Further, four-fifths of the mothers surveyed noted being concerned about where the family’s next meal would come from, highlighting the grave food insecurity in the region. Future studies should assess inflammatory and T-cell responses, treatment failure, and any subsequent progressive cardiac disease among *T. cruzi* seropositive children experiencing nutritional deficiencies to better understand the role nutritional programs could have on lowering the negative consequences of pediatric Chagas disease.

The co-existence of intestinal parasites and *T. cruzi* in endemic populations has been described [40,41]; however, very few studies have assessed the impact of concomitant *T. cruzi* and intestinal diseases. Four Spanish studies of Latin American immigrants found *Strongyloides stercoralis* serology-positive patients were significantly associated with *T. cruzi* serology positivity [42,43,44,45], highlighting the commonality of concomitant infection and the value of surveillance in adult immigrant populations in non-endemic areas. Two of these studies also found concomitant *T. cruzi* infection with *Giardia lamblia*, *Blastocystis hominis*, and *Entamoeba species* [42,43]. Interestingly, this study did not find any *Strongyloides stercoralis*-positive children by fecal qPCR, and serology was not performed due to an insufficient pediatric sample volume. Unfortunately, the underlying prevalence of strongyloidiasis in El Salvador is currently unknown, as exemplified by a 11 March 2021 literature search in PubMed, LILACS, and SciELO databases which yielded zero results.

This study is the first to demonstrate a statistically higher burden of concomitant *Giardia lamblia* and *Blastocystis hominis* in *T. cruzi* positive children. Historic animal studies have suggested that concomitant *T. cruzi* and *Strongyloides* infections can result in immunomodulating pathologic effects [46], which could impact the clinical efficacy of *T. cruzi* treatment. While the immunomodulating effect of non-*Strongyloides* intestinal parasites has not been explored, biological plausibility would reason that concomitant infection would have an immunologic altering effect. Collectively, these studies demonstrate the need to further clarify the clinical impact and long-term immunologic effects of concomitant intestinal parasitic infection in Chagas disease-positive patients. Further, future studies should evaluate toxocariasis prevalence and comorbidity impact. *Toxocara* spp. are important zoonotic nematodes with expansive geographic distribution, and the significant finding of dog ownership in this study further highlights the potential for this nematode to co-occur in Chagas disease patients. [47,48,49,50,51,52,53] Pediatric populations in particular should be investigated because of their high antiparasitic treatment response [54,55] and any potential interruption in this therapeutic effect would be of important clinical consequence.

Since 2006, a global program has been in place to target multiple human helminth infections, together with trachoma, yaws, and scabies, in an integrated program of regular and periodic preventive chemotherapy [56]. This approach now leads to the treatment of more than one billion children or adults annually and is advancing the elimination of lymphatic filariasis and trachoma, among other conditions. However, integrated antiparasitic mass treatment programs have so far excluded protozoan infections such as the ones described here. A call in 2014 to consider adding nitazoxanide or other antiprotozoal agents to mass treatment packages has so far not received serious consideration by global policymakers [57], but these and related studies might warrant a reconsideration of this aspect.

The investigative team implemented several data quality control measures to ensure optimal rigor in the study. Principally, the investigative objectives and methodologies were developed working in collaboration with local health officials and scientists in a community participatory research approach. Working with resident experts, measurement tools were developed using local focus group responses and adapting Latin American validated questionnaires [19,21]. Sonsonate’s health department provided two study nurses well-known to the community to execute data collection, and Sonsonate’s vector control provided a physical space to temporarily store biological specimens, consent forms, and questionnaires. Lastly, a formal report was provided to local partners and their buy-in was sought during manuscript development.

## 5. Conclusions

This pediatric Chagas disease surveillance investigation challenges the current dogma that “seropositive population now consists largely of persons older than the age of 50” [58]. In this study, a higher seropositive rate among children in Sonsonate, El Salvador (2.3%) was identified than previously reported for adults nationally (1.7%) [12]. This investigation identified high-burden municipalities which should be targeted for acute disease surveillance and vector control of *Triatoma dimidiata*. The use of school-based surveillance to identify neighborhood-level areas for targeted intervention is warranted and would allow for the maximization of available resources. Housing construction improvement and animal management would strengthen the sustainability of targeted interventions. Further, here a unique perspective of co-existing comorbidities is presented that will likely impact the progressive pathogenicity of pediatric cases. Concomitant intestinal parasites resulted in elevated parasite loads, and malnutrition/stunting were 4 times higher than the national average, highlighting the urgent need to consider one’s comprehensive health when treating Chagas disease pediatric cases. This study lays the foundation for future intervention efforts to improve the health and welfare of children afflicted with “the most neglected” of all neglected tropical diseases.

## Figures and Tables

**Figure 1 tropicalmed-06-00072-f001:**
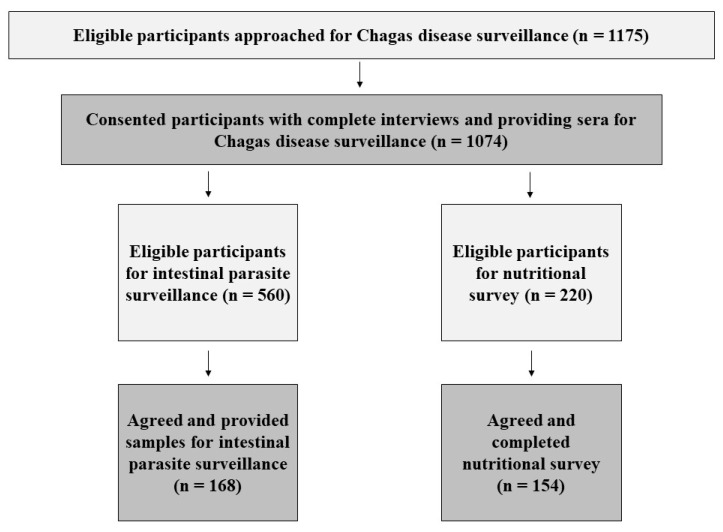
Eligibility and enrollment participant workflow of all three investigative arms.

**Figure 2 tropicalmed-06-00072-f002:**
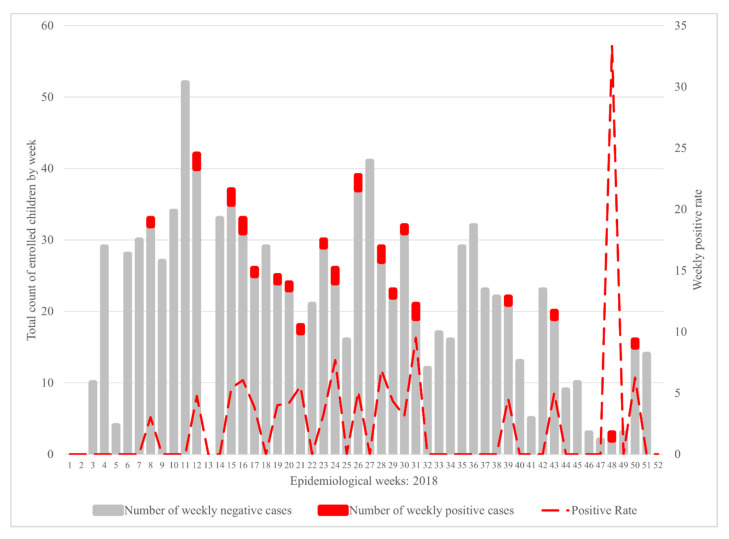
*Trypanosoma cruzi* antibody-positive pediatric cases identified throughout the calendar year 2018.

**Figure 3 tropicalmed-06-00072-f003:**
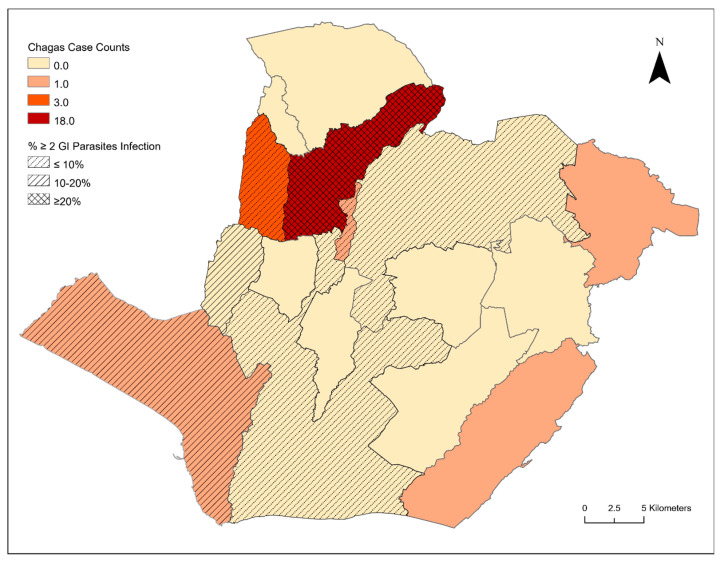
Geospatial analysis of pediatric Chagas disease and polyparasitism in Nahuizlco, Sonsonate, El Salvador.

**Figure 4 tropicalmed-06-00072-f004:**
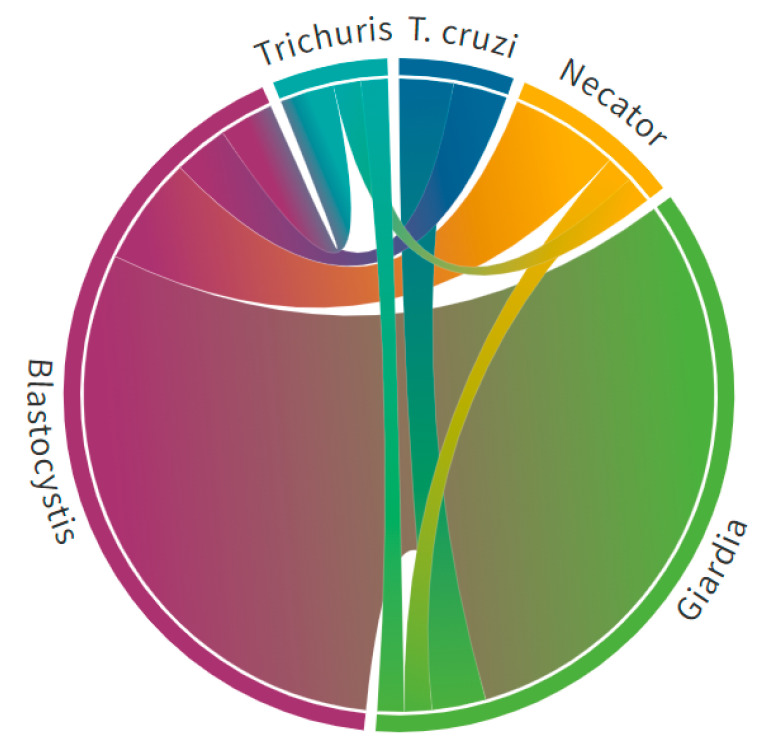
Chord diagram demonstrates a high burden of polyparasitism in 168 children from Sonsonate, El Salvador.

**Figure 5 tropicalmed-06-00072-f005:**
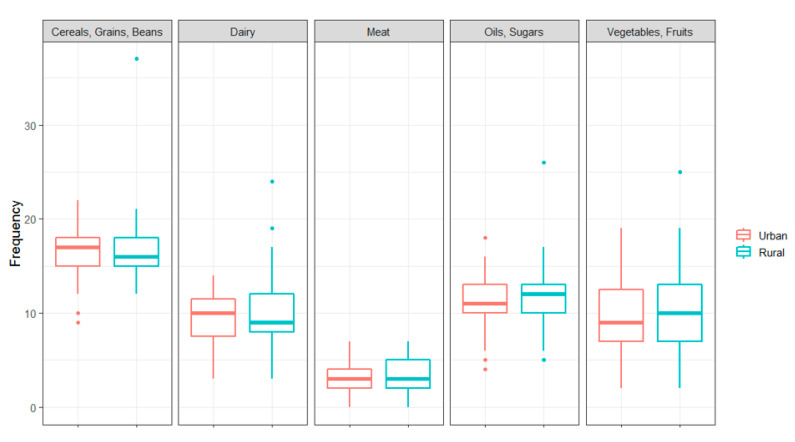
Lack of difference in food category consumption frequency noted between rural and urban households in nutrition cohort subset.

**Table 1 tropicalmed-06-00072-t001:** Risk factors associated with pediatric Chagas disease status among 1074 El Salvadorian children.

	Chagas Disease Seronegative N = 1049 (%)	Chagas Disease Seropositive N = 25 (%)	Univariate Logistic Regression Odds Ratio (95% Confidence Interval)
**Participant characteristics**			
Female	559 (53%)	11 (44%)	
Participant age (median) ^†^	8 years (range: 9 months to 18 years)	13 years (range: 5 to 18 years)	1.17 (1.07 to 1.27) *
Pediatric patient was febrile at time of enrollment	328 (31%)	8 (32%)	
Patient had other acute clinical symptoms	375 (36%)	9 (36%)	
**Household occupancy and composition**			
Number living in the household (median)	5 (range: 2 to 14)	6 (range: 4 to 31)	1.10 (1.03 to 1.17) ^¶^
Number always living in the household (median)	4 (range: 1 to 14)	5 (range: 2 to 31)	
Number of kids in household (median) ^†^	2 (range 1 to 8)	3 (range: 1 to 9)	1.36 (1.12 to 1.65) ^¶^
Number of school-aged children	1 (range 0 to 5)	2 (range: 0 to 6)	1.76 (1.32 to 2.35) *
Number of women of childbearing age (median)	1 (range: 0 to 5)	1 (range: 0 to 7)	
Number of currently pregnant women (median)	0 (range: 0 to 1)	0 (range: 0 to 1)	
Mother’s last year of education (median)	5 (range: 0 to 15)	2 (range: 0 to 12)	0.82 (0.72 to 0.94) ^¶^
Father’s last year of education (median)	6 (range: 0 to 15)	5 (range: 0 to 8)	0.89 (0.80 to 0.99) ^§^
**Household poverty indicators**			
Agriculture as primary household income source	613/966 (63%)	16 (64%)	
Household receives remesas	59 (6%)	1 (4%)	
Number of beds in home (median)	4 (range: 2 to 14)	5 (range: 0 to 16)	1.19 (1.05 to 1.34) ^¶^
Bare earth floor in bedroom	820 (78%)	20 (80%)	
Adobe wall material in bedroom	443/1047 (42%)	14 (56%)	
Leña cooking fuel type	145/1047 (14%)	4 (16%)	
Potable water in the house	919 (88%)	22 (88%)	
Family uses outdoor latrine for bathroom	961 (92%)	25 (100%)	
House has electricity	656/1035 (64%)	12 (48%)	
House has fowl (chickens or turkeys)	544 (52%)	14/24 (58%)	
House has dogs ^†^	533/1011 (53%)	19/24 (79%)	3.41 (1.27 to 9.20) ^§^
House has fowl and dogs	281/1011 (28%)	9/24 (37.5%)	
**Household triatomine exposure and parent or legal guardian Chagas disease knowledge**			
Knows what a “chinche” is	716 (68%)	15 (60%)	
Has seen chinches inside their house within the past year	295/716 (41%)	9/15 (60%)	
Someone in house has been bitten by a chinche in the past year	143/714 (20%)	3/15 (20%)	
Your house has been fumigated in past year	107/1047 (10%)	1/25 (4%)	
Knows about Chagas disease	200/1047 (19%)	4/25 (16%)	
Know someone that has or had Chagas disease	43/200 (22%)	0/4 (0%)	
**Other household health concerns**			
Someone in your house sought clinical care in the past year	714/1027 (70%)	15/23 (65%)	
Household member(s) who sought clinic care in the past year went for an infection origin	404/714 (57%)	10/15 (67%)	
Someone in your household took medicine for intestinal parasites in the past year	732 (69%)	21/25 (84%)	

Variable significant on univariate regression are as follows: ^§^
*p*-value < 0.05; ^¶^
*p*-value < 0.01; * *p*-value < 0.001. ^†^ Variable was significant on final multivariate regression, *p*-value < 0.05.

**Table 2 tropicalmed-06-00072-t002:** Nutritional deficiencies identified in nutritional cohort subset, stratified by urban versus rural living within the state of Sonsonate, El Salvador.

	Urban N = 58 (%)	Rural N = 96 (%)	Univariate *p*-Value OR (95% CI)
***Child’s health status and concerns***			
Child’s age at time of survey (median)	9 years (range: 1–18 years)	8 years (range: 1–18 years)	0.374
Child’s mother reported prenatal complications	5 (9%)	12 (13%)	0.480
Child was breastfed	53 (92%)	85 (89%)	0.577
Length mother breastfed (median)	16.5 months (range: 2–48 months)	17.5 months (range: 2–48 months)	0.996
Number of times child was sick last year (median)	0 (range: 0–4 times)	0 (range: 0–4 times)	0.812
Child was febrile at time of survey	14 (24%)	7 (7%)	0.005 0.25 (0.09–0.66)
Child was Chagas disease seropositive	2 (3%)	1 (1%)	0.332
Child with suspected GI parasitic infection	11 (19%)	19 (20%)	0.586
Mother has health concerns for child	12 (21%)	18 (19%)	0.768
Mother has growth and development concerns for child	3 (5%)	3 (3%)	0.529
Mother has educational concerns for child	6 (10%)	14 (15%)	0.450
Mother has safety concerns for child	1 (2%)	4 (4%)	0.422
***Household and participant’s hygienic practices***			
Dirt floor in child’s bedroom	39 (67%)	59 (61%)	0.546
Child washes hands with soap before eating *	58 (100%)	91 (95%)	-
Child washes hands with soap after restroom *	58 (100%)	94 (98%)	-
Child bathes 3+ times per week	58 (100%)	93 (97%)	0.743
Household uses a refrigerator for food storage	20 (34%)	42 (44%)	0.694
Household uses well water for drinking *	0 (0%)	55 (57%)	-
Household uses well water for cooking *	0 (0%)	56 (58%)	-
***Household nutrition and food insecurity***			
Family eats the same thing each day	37 (64%)	58 (60%)	0.860
Majority of consumed food is grown at home	5 (9%)	26 (27%)	0.011 3.77 (1.35–10.51)
Mother familiar with nutritional guidelines	9 (15%)	2 (2%)	0.012 0.13 (0.03–0.64)
Child went 1+ day(s) without a meal last week	4 (7%)	7 (7%)	0.022 0.20 (0.05–0.79)
Family went 1+ day(s) without a meal last month	3 (5%)	13 (14%)	0.438
Mother worries about family’s next meal	37 (64%)	75 (78%)	0.072

* Zero count cells yielded no statistic results.

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
