# Peer review of "Elevated Pediatric Chagas Disease Burden Complicated by Concomitant Intestinal Parasites and Malnutrition in El Salvador"

_tropicalmed, 2021, doi:10.3390/tropicalmed6020072_

Round 1

Reviewer 1 Report

This ms describes a study aiming at evaluating pediatric T. cruzi seroprevalence investigation with a simultaneous study arms assessing gastrointestinal parasite concomitant infection and malnutrition in a region of El Salvador. The topic is important, the study has been carried out properly and the ms is well presented. I have only few minor suggestions for the Authors before the ms can be considered suitable for publication.

General Comment

The investigated correlation between Trypanosoma cruzi seroprevalence and occurrence of intestinal human parasite is truly important. Nonetheless, considering the availability of blood from children, other data of high importance would have come from the evaluation of seropositivity to the most important soil-transmitted zoonotic nematode, i.e. Toxocara causing larva migrans syndromes, which I expect to be high in settings like those which have been investigated. Of course I am not asking the Authors to perform such additional analysis, but at least they should add some discussion on this topic, leaving a space open for the future. In particular, this small section should briefly

  1. a) remind the clinical impact of this canine and feline parasite in humans. See and cite this ref:

https://pubmed.ncbi.nlm.nih.gov/23305972/

  1. b) summarize the importance of the worldwide environmental contamination by Toxocara eggs, not only in rural and developing settings (See and cite this exeplificative ref: https://pubmed.ncbi.nlm.nih.gov/18949345/) but also in urbanized public green areas (see and cite these refs: https://pubmed.ncbi.nlm.nih.gov/31627164/; https://pubmed.ncbi.nlm.nih.gov/32369482/; https://pubmed.ncbi.nlm.nih.gov/18820328/), as in both cases the ingestion of these eggs represent a major threat for human health, especially in children.
  2. c) accordingly, a mention that there is a high percentage of infection by zoonotic roundworms in canine populations all over the World, in both developing and developed, rural and urban, settings, would be of benefit. Some references to discuss and cite as examples are https://pubmed.ncbi.nlm.nih.gov/32516309/; https://pubmed.ncbi.nlm.nih.gov/23538502/; https://pubmed.ncbi.nlm.nih.gov/33308762/; https://pubmed.ncbi.nlm.nih.gov/31067231/; https://pubmed.ncbi.nlm.nih.gov/31014530/)

Minor comments

1) Reword sentences of the ms using “we” or “our”, and rather use an impersonal style

2) The investigated parasites localize in the bowel and not also in the stomach so change the term “gastrointestinal” into “intestinal” throughout the ms

3) Line 41, delete “(T. cruzi) parasite”

4) Line 120: these are nine parasites, not eight as stated, and the bracket should not be in italics

Author Response

Reviewer One:

This ms describes a study aiming at evaluating pediatric T. cruzi seroprevalence investigation with a simultaneous study arms assessing gastrointestinal parasite concomitant infection and malnutrition in a region of El Salvador. The topic is important, the study has been carried out properly and the ms is well presented. I have only few minor suggestions for the Authors before the ms can be considered suitable for publication.

General Comment

The investigated correlation between Trypanosoma cruzi seroprevalence and occurrence of intestinal human parasite is truly important. Nonetheless, considering the availability of blood from children, other data of high importance would have come from the evaluation of seropositivity to the most important soil-transmitted zoonotic nematode, i.e. Toxocara causing larva migrans syndromes, which I expect to be high in settings like those which have been investigated. Of course I am not asking the Authors to perform such additional analysis, but at least they should add some discussion on this topic, leaving a space open for the future. In particular, this small section should briefly

  1. a) remind the clinical impact of this canine and feline parasite in humans. See and cite this ref:

https://pubmed.ncbi.nlm.nih.gov/23305972/

  1. b) summarize the importance of the worldwide environmental contamination by Toxocara eggs, not only in rural and developing settings (See and cite this exeplificative ref: https://pubmed.ncbi.nlm.nih.gov/18949345/) but also in urbanized public green areas (see and cite these refs: https://pubmed.ncbi.nlm.nih.gov/31627164/; https://pubmed.ncbi.nlm.nih.gov/32369482/; https://pubmed.ncbi.nlm.nih.gov/18820328/), as in both cases the ingestion of these eggs represent a major threat for human health, especially in children.
  2. c) accordingly, a mention that there is a high percentage of infection by zoonotic roundworms in canine populations all over the World, in both developing and developed, rural and urban, settings, would be of benefit. Some references to discuss and cite as examples are https://pubmed.ncbi.nlm.nih.gov/32516309/; https://pubmed.ncbi.nlm.nih.gov/23538502/; https://pubmed.ncbi.nlm.nih.gov/33308762/; https://pubmed.ncbi.nlm.nih.gov/31067231/; https://pubmed.ncbi.nlm.nih.gov/31014530/)

Thank you, we have added a section regarding this limitation and rationale for its future study in this context (lines 383-387). We have included these recommended references. We do agree with the reviewer, and will be sure to consider this additional and important parasite in future investigations.  

 Minor comments

  • Reword sentences of the ms using “we” or “our”, and rather use an impersonal style

Thank you, we have revised all pronouns accordingly.

  • The investigated parasites localize in the bowel and not also in the stomach so change the term “gastrointestinal” into “intestinal” throughout the ms

Thank you, we have revised all nouns as recommended.

  • Line 41, delete “(T. cruzi) parasite”

Thank you, we have revised accordingly.

4) Line 120: these are nine parasites, not eight as stated, and the bracket should not be in italics

Thank you, we have corrected this error.

Reviewer 2 Report

The study took place in the State of Sonsonate, El Salvador. There, among the principal causes of mortality, are infectious and parasitic disieases, including Chagas disease. The dominant insect vector is Rhodnius prolixus which was eradicated in 2010. Since 2011 exists a gap in the data to estimate whether new insect vector species have been established in the region. So, in the present manuscript the authors analyses, in a cross section study, seroprevalence associated to two possible risks factors: gastrointestinal parasites and malnutrition. 

The focus of the study is good because, as the authors mention, few reports analysed Chagas disease associated with other clinical commorbidities. It had been reported that bad nourished children can carried parasite infections and in Latin America most of these areas are endemic for Chagas disease.

The originality of the study is that authors look for seroprevalence in an area where the usual vector was eradicated and made their cross section study between positive serology and two risk factors. They registered that significant association with parasitic infection depends in the parasitic load. In general, the study was well designed and performed, and results are very well presented.

Author Response

The study took place in the State of Sonsonate, El Salvador. There, among the principal causes of mortality, are infectious and parasitic disieases, including Chagas disease. The dominant insect vector is Rhodnius prolixus which was eradicated in 2010. Since 2011 exists a gap in the data to estimate whether new insect vector species have been established in the region. So, in the present manuscript the authors analyses, in a cross section study, seroprevalence associated to two possible risks factors: gastrointestinal parasites and malnutrition. 

The focus of the study is good because, as the authors mention, few reports analysed Chagas disease associated with other clinical commorbidities. It had been reported that bad nourished children can carried parasite infections and in Latin America most of these areas are endemic for Chagas disease.

The originality of the study is that authors look for seroprevalence in an area where the usual vector was eradicated and made their cross section study between positive serology and two risk factors. They registered that significant association with parasitic infection depends in the parasitic load. In general, the study was well designed and performed, and results are very well presented.

Author response: Thank you for the kind review.

Reviewer 3 Report

Dear Editor,

I carefully read the article by Nolan et al., which is overall interesting.

My comments and suggestions for the authors are the following:

  • Why did the authors choose to include in the multivariate analysis also non-significant predictors in the univeriate analysis (i.e. any univariate variables with p<0.25)? I think that including only univariate variable with p-value< 0.05 is more formally appropriate.
  • Table 1 is explicative. However, all the used abbreviations should be defined at the bottom of the table.
  • In tables 1 and 2, P-values are unnecessary when OR is reported along with its 95%CI. Non-significant P-values are unnecessarily reported too. Then, I suggest the authors to remove P-values in order to improve the readability of the table.
  • The limitations of the study need to be deeper discussed in the manuscript.
  • English language needs to be carefully revised.

Author Response

Reviewer 3:

I carefully read the article by Nolan et al., which is overall interesting.

My comments and suggestions for the authors are the following:

  • Why did the authors choose to include in the multivariate analysis also non-significant predictors in the univariate analysis (i.e. any univariate variables with p<0.25)? I think that including only univariate variable with p-value< 0.05 is more formally appropriate.
  • Table 1 is explicative. However, all the used abbreviations should be defined at the bottom of the table.
  • In tables 1 and 2, P-values are unnecessary when OR is reported along with its 95%CI. Non-significant P-values are unnecessarily reported too. Then, I suggest the authors to remove P-values in order to improve the readability of the table.
  • The limitations of the study need to be deeper discussed in the manuscript.
  • English language needs to be carefully revised.

Author response: Thank you for your review. We originally used a p<0.25 cut-off to accommodate for a potential influence of lower power subsequent from small sample size (n=25 chagas positive children). We have updated our multivariate analysis calculations and methods statement as recommended. We believe table 1 has appropriate footnotes. Please let us know what specific additional abbreviations need clarification, and we will be happy to correct. We have revised Tables 1 and 2 as recommended, with footnotes that correspond to significance level: § p-value <0.05; ¶ p-value <0.01; * p-value <0.001. We have added additional limitations sentences throughout the discussion, where they apply to each topical focus area. Finally, we have reviewed the manuscript again and made grammatical/English language improvements.  

Round 2

Reviewer 3 Report

Dear Editor,

I carefully read the revised version of the manuscript. I have still some concerns regarding the manuscript:

  • Authors roughly revised the tables. As a matter of fact, they did not even notice they did not defined some used abbreviations (i.e. "N", "OR" and "CI").
  • They should include a post-hoc power test for the significant main outcomes, in order to evaluate if their analysis is powered or if it is not.

Author Response

Thank you bringing to light a potential limitation. To address the reviewer's concern, we have taken two actions: 1) performed a sample size and power analysis, and 2) added a sentence as to the potential limitation of overall number of Chagas seropositive children identified (lines 345-346). Using EpiInfo v7.2''s Stat-Calc program (CDC, Atlanta, GA), we performed a cross-sectional sample size and power estimate using the following variables: seroprevalence of 2.3%, 0.75% margin of error, pediatric population in Sonsonate of 148,160, and a design effect of 1. These calculations resulted in a needed sample size of 653 to achieve a 80% power and 1,073 sample size for 90% power. Given our final sample size was 1,074, we anticipate a 90% power in our surveillance findings. While this is above the acceptable scientific limit (>80% power), we acknowledge that 90% is not an ideal power level. Therefor, as aforementioned, we have added this limitation to the discussion section. We hope this satisfies the reviewer's concern.